# Effective Macrosomia Prediction Using Random Forest Algorithm

**DOI:** 10.3390/ijerph19063245

**Published:** 2022-03-10

**Authors:** Fangyi Wang, Yongchao Wang, Xiaokang Ji, Zhiping Wang

**Affiliations:** 1Department of Occupational and Environmental Health, School of Public Health, Cheeloo College of Medicine, Shandong University, Jinan 250012, China; 201935825@mail.sdu.edu.cn; 2Department of Biostatistics, School of Public Health, Cheeloo College of Medicine, Shandong University, Jinan 250012, China; wych@sdu.edu.cn (Y.W.); jxk@sdu.edu.cn (X.J.)

**Keywords:** random forest, macrosomia, interspinal diameter, sacral external diameter, transverse outlet

## Abstract

(1) Background: Macrosomia is prevalent in China and worldwide. The current method of predicting macrosomia is ultrasonography. We aimed to develop new predictive models for recognizing macrosomia using a random forest model to improve the sensitivity and specificity of macrosomia prediction; (2) Methods: Based on the Shandong Multi-Center Healthcare Big Data Platform, we collected the prenatal examination and delivery data from June 2017 to May 2018 in Jinan, including the macrosomia and normal-weight newborns. We constructed a random forest model and a logistic regression model for predicting macrosomia. We compared the validity and predictive value of these two methods and the traditional method; (3) Results: 405 macrosomia cases and 3855 normal-weight newborns fit the selection criteria and 405 pairs of macrosomia and control cases were brought into the random forest model and logistic regression model. On the basis of the average decrease of the Gini coefficient, the order of influencing factors was: interspinal diameter, transverse outlet, intercristal diameter, sacral external diameter, pre-pregnancy body mass index, age, the number of pregnancies, and the parity. The sensitivity, specificity, and area under curve were 91.7%, 91.7%, and 95.3% for the random forest model, and 56.2%, 82.6%, and 72.0% for logistic regression model, respectively; the sensitivity and specificity were 29.6% and 97.5% for the ultrasound; (4) Conclusions: A random forest model based on the maternal information can be used to predict macrosomia accurately during pregnancy, which provides a scientific basis for developing rapid screening and diagnosis tools for macrosomia.

## 1. Introduction

Macrosomia refers to live-born newborns with birth weight ≥ 4000 g, which is one of the adverse outcomes of newborns [1]. Macrosomia increases the rate of a cesarean section and several maternal and newborn complications [2]. These complications are associated with increased risks of shoulder dystocia, brachial plexus injury, asphyxia, prolonged labor, postpartum hemorrhage, and laceration of the anal sphincter [3]. As a consequence, macrosomia infants, on average, have higher health care utilization [4]. It is of great significance for obstetricians to predict macrosomia in the early stage of pregnancy, and it is helpful to have a timely intervention for puerperal, to reduce the injury of puerperal and newborn, to reduce the incidence of complications during delivery, and to improve the quality of obstetrics.

At present, the occurrence of macrosomia is related to factors such as genes [5], pre-pregnancy body mass index [6], excessive weight gain during pregnancy [7], and gestational diabetes mellitus [8], but few studies have built a high-accuracy prediction model based on these factors. At present, clinical workers often use the Hadlock formula built into ultrasonic instruments to predict fetal body mass [9]. However, previous studies have shown that the accuracy of ultrasonic diagnosis of macrosomia is low (sensitivity: 12–75%, specificity: 68–99%). A low sensitivity may result in much macrosomia being missed, increasing the incidence of complications in both the mother and the newborn [10]. Therefore, we wanted to build a prediction model to deal with this problem. In the aspect of model construction, traditional prediction of macrosomia mainly used the logistic regression model or the Cox proportional hazards regression model, but the predictive efficiency is not very high [11]. It is urgent to introduce advanced methods into the construction of the macrosomia prediction model. Random forest is a methodology for learning from existing data to make predictions on new data [12]. Random forest improves the prediction accuracy without significantly increasing the amount of computation, and it is not sensitive to multicollinearity [13]. It can reasonably predict the role of up to thousands of explanatory variables and is known as one of the best algorithms at present [14]. However, its use in the prediction of macrosomia has been relatively limited. Random forest models have been used for many medical predictions; one study used pre-pregnancy and 26-week follow-up data to predict large-for-gestational-age infants, with an area under curve of 0.824 [4].

Therefore, if we can identify women in early pregnancy with the tendency to deliver macrosomia according to some characteristics and carry out the proper pregnancy guidance, it will help to reduce the incidence of macrosomia. Based on the Shandong Multi-Center Healthcare Big Data Platform, this study applied the random forest algorithm to construct an auxiliary prediction model for macrosomia to identify the risk factors of macrosomia, and the prediction ability was compared with that of traditional logistic regression model and ultrasound, providing a reference for the development of macrosomia prevention and early intervention strategies.

## 2. Materials and Methods

### 2.1. Data Sources and the Subject

A community-based cohort was created from the “Shandong Multi-Center Healthcare Big Data Platform” (SMCHBDP) in China, and the prenatal examination data in outpatient records and the delivery data in hospital delivery records of the pregnant women who delivered newborns in Jinan city from June 2017 to May 2018 were collected. The information collected included: (a) newborns’ birth weight in the hospital delivery records; (b) estimated weight of newborns using the Hadlock formula built into the ultrasonic instruments; (c) mother’s basic demographic information in outpatient records, such as age, pre-pregnancy BMI, number of pregnancies, and parity; (d) extrapelvic measurement information in the first prenatal examination records at the twelfth week of pregnancy, including interspinal diameter, intercristal diameter, sacral external diameter, and transverse outlet. 

Newborns were our study objects in which macrosomia were the cases and the normal-weight newborns were the controls. The inclusion criteria of the subjects were: (a) newborn birth, weight ≥ 2500 g with singleton and live birth; (b) permanent address of the mother is in Jinan; (c) both the hospital for the maternal pregnancy examination and for delivery are in Jinan; (d) with complete explanatory variable information needed for this study. The inclusion criterion for the case was the weight of the newborn ≥ 4000 g and for the control was the weight ≥ 2500 g and <4000 g. Previous studies have confirmed that data imbalance will have a negative impact on the accuracy of random forest model classification results, and most medical data are unbalanced data. The incidence of macrosomia is much lower than the non-incidence, so the number of the subjects obtained in the two groups was unbalanced. To avoid the false high accuracy caused by the bias of the classifier to most classes, this study adopted under-sampling technology. It selected the same number of the normal-weight newborns as the case group according to the ratio of 1:1 as the control [15]. All macrosomia that met the criteria were included in the study. We used the improved under-sampling method to extract the control, it repeatedly sampled and trained multiple classifiers, which not only avoided the loss of key information but also alleviated the over-fitting [16]. 

### 2.2. The Construction of Prediction Models

In this model, the outcome variable was whether the subjects presented with macrosomia, which was the goal of the decision-making classification. The explanatory variables were multiple risk factors that may cause macrosomia, such as age, pre-pregnancy BMI, the number of pregnancies, parity, and extrapelvic measurement. They were used to classify the outcome variables. The subjects in the case and control groups were randomly divided into the training set and the test set according to the percentage of 7:3, and the two datasets were independent [17].

Firstly, the subjects were brought into the training set, modeled by the random forest algorithm and the logistic regression model. A stepwise regression was used in the logistic regression model, and the inclusion and exclusion indices of the model were SLE = SLS = 0.15. A VIF index was used to judge whether there was multicollinearity between the variables. In random forest model construction, two parameters will mainly affect its work efficiency: [18] (1) the number of trees (ntree) and (2) the candidate feature subset (mtry). Therefore, selecting appropriate parameters can ensure the stability of the model. 

Secondly, the two models were tested in the test set, and the area under the receiver operating characteristic curve was measured to evaluate the accuracy of the model in predicting the occurrence of macrosomia. The sensitivity, specificity, positive predictive value, negative predictive value, and other indicators were obtained. The above indices were compared with the B-ultrasound. The importance of explanatory variables in the development of outcomes can be evaluated. The Gini coefficient evaluates the influence of each explanatory variable on the heterogeneity of observations at each node in the decision tree. The more significant the average decline of the Gini coefficient, the more critical the explanatory variable is for classification to obtain the importance ranking.

Thirdly, the predicted weight obtained from the last prenatal B-ultrasound examination was compared with the gold standard for whether actual birth weight ≥ 4000g . The predicted weight ≥ 4000 g was denoted as 1, and the weight < 4000 g was denoted as 0, thus obtaining the accuracy of the prediction of macrosomia by the last prenatal B-ultrasound examination. 

At last, we compared the accuracy and predictive value of the three methods with ultrasonography.

Because of the use of under-sampling method, some information may have been lost in the analysis. We cross-validated external data to confirm the stability of the model.

### 2.3. Statistical Analysis

Categorical variables were described as absolute values and percentages, and continuous variables were expressed as means ± standard deviations or medians and interquartile ranges. The mean difference was assessed by Student’s *t*-test, and the median difference was assessed by the Mann–Whitney test. All analyses were performed using R software version 4.0.2 (R Foundation for Statistical Computing, Vienna, Austria; http://www.r-project.org, accessed on 25 February 2022). *p* value < 0.05 was considered significant. We used the “randomForest “package to develop the random forest models, and the” ROSE “package was used to sort imbalanced data. The “MASS “package was used to build the logistic regression model.

## 3. Results

### 3.1. Essential Population Characteristics

In this study, we firstly collected 4260 newborns that fit the criteria of the subjects, in which 405 were macrosomia and 3855 were normal-weight newborns. The prevalence of macrosomia based on the big data platform was 9.5% in the newborn for singleton and live birth. Compared with the population delivering normal-weight newborns, the mothers who delivered macrosomia newborns had higher age, pre-pregnancy BMI and interspinal diameter and a lower sacral external diameter. The number of pregnancies and parity were different between the two groups (Table 1). There was no significant difference in mothers’ information between the control selected by the under-sampling method and total normal weight newborns (Table 1), indicating that the randomly selected control had good representativeness.

### 3.2. Macrosomia Establishment of Random Forest Model

Based on the above information, a random forest model was established. A total of 405 normal-weight newborns were matched by a 1:1 under-sampling technique as control, and a total of 810 subjects were brought into the model. When the number of random seeds was 666, the number of candidate feature subsets (mtry) was 4, and the number of fixed decision trees (ntree) was 1–500, and the variation of the average out-of-bag estimation error rate with ntree was observed (Figure 1). When ntree was 1–50, the average out-of-bag estimation error rate decreased rapidly, but it decreased slowly after 50 and tended to be stable after 500. Therefore, this study selected the number of decision trees when ntree was 500 to obtain the optimal model. The overall misjudgment rate of the model based on the OBB data was 6.34%. In line with the average decline of the Gini coefficient of each risk factor in the random forest model, the importance ranking of explanatory variables were obtained (Figure 2). The predictive macrosomia factors that were screened by the variable importance measure in the random forest were obtained as follows: interspinal diameter, transverse outlet, intercristal diameter, sacral external diameter, pre-pregnancy BMI, age, the number of pregnancies, and the parity. The number of decision tree nodes in the model was at least 45 and at most 85 (Figure 3). The classification accuracy was 92.6%, the sensitivity was 88.4%, and the specificity was 96.7% (Figure 4). We conducted cross-validation 9 times with the remaining 3450 unselected controls, and the results were relatively robust, as shown in Figure 5.

### 3.3. The Comparison of the Three Methods in Predicting Macrosomia

According to the predicted weight obtained from the last prenatal ultrasound examination, the median gestational age of macrosomia at the last ultrasound was 39.9 weeks and the interquartile range was 1.5 weeks. The median gestational age at the last ultrasonography of normal weight infants was 39.1 weeks, and the interquartile interval was 1.6 weeks. In total, 127 cases were foreseen as macrosomia, and 278 cases were predicted as normal-weight newborns of 405 macrosomia newborns. Conversely, 3762 controls were predicted as normal-weight newborns and 93 controls were predicted as macrosomia newborns of 3855 normal-weight newborns. As shown in Table 2, the AUC, sensitivity, specificity, Youden’s index, false-negative rate, false-positive rate, positive predictive value, and negative predictive value of the random forest model were 0.953, 91.7%, 91.7%, 83.4%, 8.3%, 8.3%, 91.7%, and 91.7%. The AUC, sensitivity, specificity, Youden’s index, false-negative rate, false-positive rate, positive predictive value, and negative predictive value of the logistic regression model were 0.720, 52.6%, 82.6%, 38.8%, 43.8%, 17.4%, 70.8%, and 37.9%. The sensitivity, specificity, Youden’s index, false-negative rate, false-positive rate, positive predictive value, and negative predictive value of ultrasound in predicting macrosomia were 29.6%, 97.6%, 27.2%, 70.1%, 2.4%, 57.7%, and 93.1%. Therefore, we can infer that the current value of ultrasound in predicting macrosomia is low, and a more accurate model is needed. The comparison of the three methods is shown in Figure 4.

## 4. Discussion

Previous studies on macrosomia were limited to screening risk factors for macrosomia and could not be used to predict macrosomia [19]. Therefore, the most useful method to estimate fetal body weight at present is ultrasonography. There are many methods to predict newborn birth weight, such as the Hadlock formula [20], Merz E formula [21], Ott WJ formula [22], Combs CA formula [23], and Scioscia M formula [24], etc. The most common used formula in China is the Hadlock formula, which is based on four fetal biological indices: abdominal circumference (AC), femoral neck (FL), biparietal diameter (BPD), and head circumference (HC). However, a review of 14 studies showed widely varying diagnostic results for the sonographic detection of macrosomia (≥4000 g) in general obstetric populations (sensitivity: 12–75%, specificity: 68–99%, post-test probability: 17–79%). Moreover, in this study, we calculated the accuracy of ultrasound in predicting macrosomia. We used the last B-ultrasound before delivery for prediction. The sensitivity was 29.6%, and the specificity was 97.6%. Previous studies indicated that the B-ultrasound at 35th week of gestational age may have greater prediction than the last B-ultrasound before delivery for prediction [25], so the low discrimination may be related to the proximity to delivery.

As a machine learning algorithm, a random forest algorithm can value the importance of a variable for the prediction of dependent variables and can provide a reference basis for subsequent clinical decision-making. Previous studies on the risk factors of macrosomia mainly used the traditional logistic regression model or Cox proportional hazards regression model. These models have special requirements for data distribution and are sensitive to multivariate collinearity, so they have some limitations in application. However, a random forest algorithm can overcome these limits [26]. The sensitivity, specificity, and AUC of the logistic regression model in this study were 56.2%, 82.6%, and 72.0%. The predictive power is indeed lower than that of random forest. Random forest is widely used in the medical area, and it can be used to distinguish and classify gene–gene expression data [27] and protein action sites [28]. In terms of application of the model prediction, a study in China used random forest to predict heatstroke [29]. Another study established a weather-based forecasting and early warning model for HFMD [30], with the sensitivity and specificity higher than 0.90. However, the use of a random forest model to predict macrosomia is rare in obstetrics and gynecology. In this paper, we show that building random forest model to predict the occurrence of macrosomia is better than ultrasound. The specificity of ultrasound in predicting macrosomia is very high, indicating that the false-positive rate is low. However, random forest significantly improves the sensitivity from 29.6% to 91.7%, improves the positive predictive value of predicting macrosomia, reduces the false-negative rate, has great significance for macrosomia prevention, and significantly improves the predictive value. The specificity and false positive rate between two methods are close. 

The under-sampling method eliminates class imbalance by reducing the number of samples in most classes, the simplest and most effective under-sampling method is Random under-sampling (RUS), which is consistent with the idea of random sampling. This method randomly selects samples of the same size as those of the minority samples from the majority samples and then combines the selected samples with the minority samples to form a new balanced sample set. Obviously, selecting some samples randomly from the population will cause the loss of key information in the population. Based on this problem, we used the improved under-sampling method and used the EasyEnsemble algorithm to repeatedly sample most classes and train multiple classifiers, which not only avoids the problem of key information loss caused by under-sampling but also alleviates the problem of over-fitting, making the sample more representative.

The strengths of the study are the early use of a big data platform in China and building an early prediction model for macrosomia. Our study was limited by the lack of validation of data from other regions other than Jinan city. In addition, a large number of mothers with missing basic information were excluded. A future study should bring more data from more regions into the model. During the first prenatal examination in the early stage of pregnancy, the tendency of pregnant women to produce macrosomia can be predicted, so that the pregnant women with low fetal weight predicted by B-ultrasound can be more alert, and the occurrence of emergencies during labor can be reduced. Extrapelvic measurements are a standard method to estimate whether spontaneous labor is normal or not. Using them to estimate macrosomia has a specific new significance. We can obtain the possible results of prediction in the early stage of pregnancy, which also has a distinct guiding sense for pregnancy health care.

## 5. Conclusions

The random forest algorithm identified the pelvis as an essential factor in predicting the occurrence of macrosomia from the basic information of mothers in pregnancy, which provided a basis for the prevention and early intervention of macrosomia. In conclusion, a random forest model based on the maternal index can be used to diagnose macrosomia accurately during pregnancy and provide a scientific basis for developing rapid screening and diagnosis tools.

## Figures and Tables

**Figure 1 ijerph-19-03245-f001:**
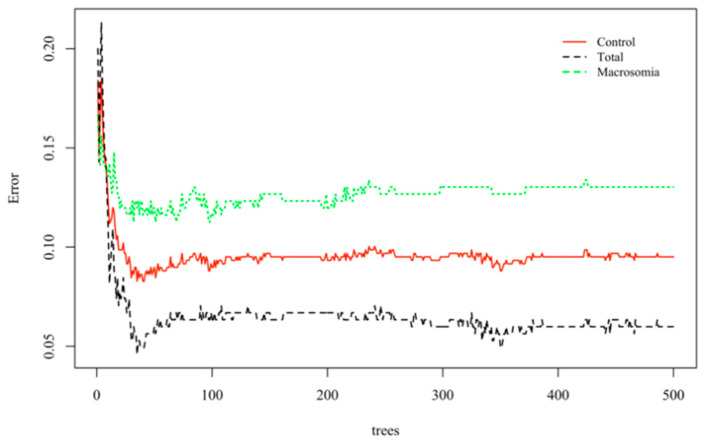
Change diagram of the number of decision trees and the average out-of-bag estimated error rate when establishing a random forest model.

**Figure 2 ijerph-19-03245-f002:**
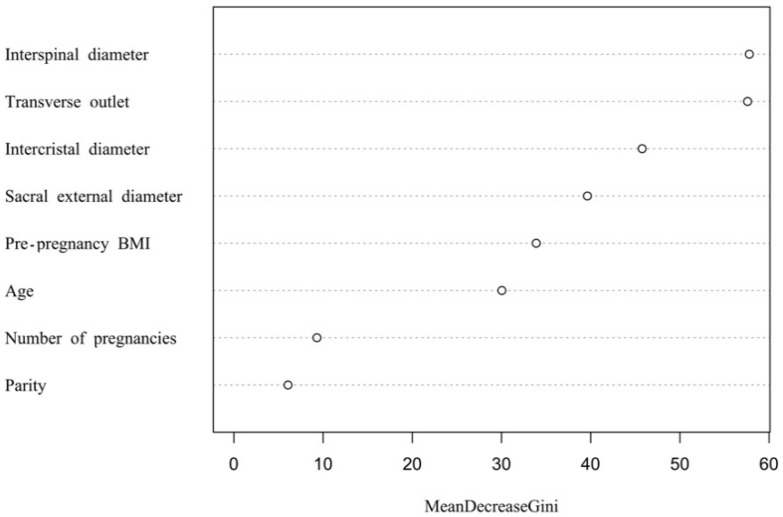
The figure of variable importance ranking in the macrosomia random forest prediction model.

**Figure 3 ijerph-19-03245-f003:**
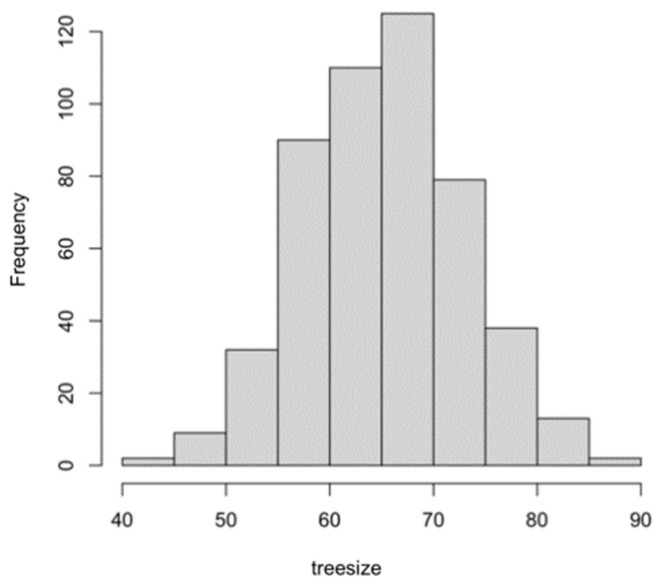
The histogram of treesize of the random forest model.

**Figure 4 ijerph-19-03245-f004:**
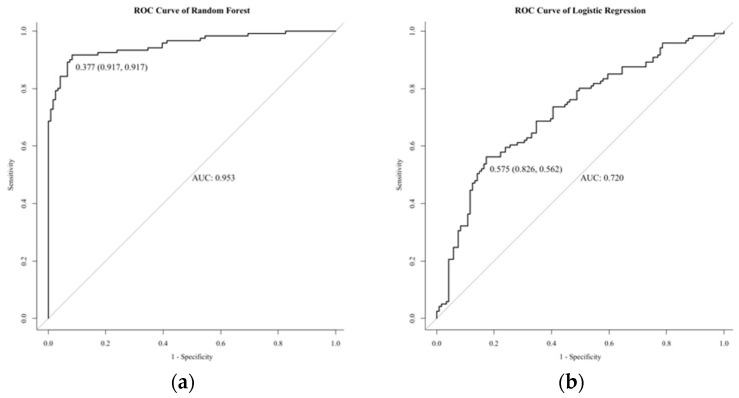
The receiver operating characteristic curve of the three methods. (**a**) shows the ROC curve of the random forest model in predicting macrosomia. (**b**) shows the ROC curve of the logistic regression in predicting macrosomia.

**Figure 5 ijerph-19-03245-f005:**
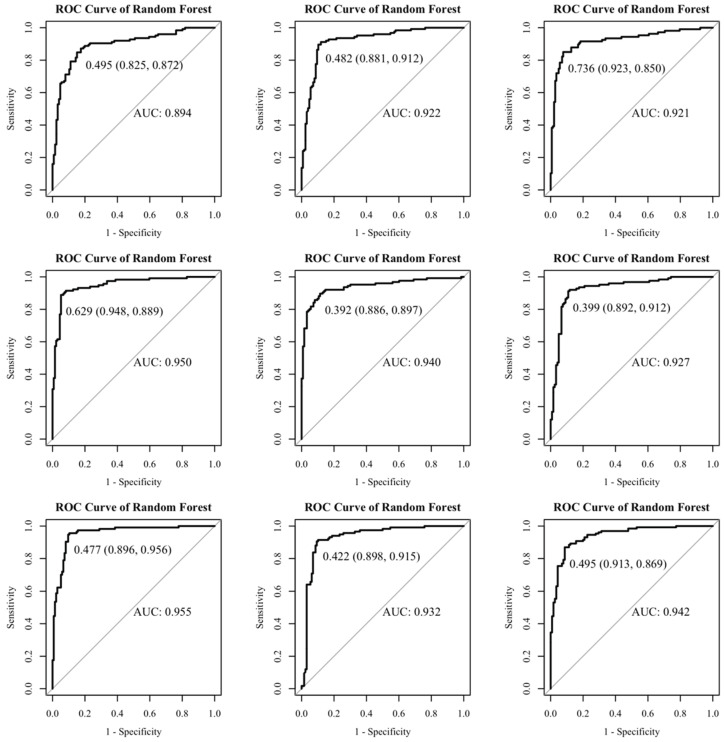
The cross-validation results.

**Table 1 ijerph-19-03245-t001:** Characterization of macrosomia and control groups and the representativeness of the control.

Variables	Total Subjects			Control		
	Macrosomia (N = 405) (A)	Normal Weight Newborns (N = 3855) (B)	*p*	Not Selected (N = 3450) (C)	Selected (N = 405) (D)	*p*
Age (years old)	30.2 ± 4.2	28.1 ± 3.7	<0.001	28.9 ± 3.8	29.2 ± 4.1	0.251
Pre-pregnancy BMI (Kg/m^2^)	23.6 ± 4.0	22.3 ± 3.2	<0.001	21.9 ± 3.1	22.1 ± 3.3	0.236
Gestational Age (week)	40.1 ± 0.9	39.8 ± 1.1	0.285	39.8 ± 1.1	39.9 ± 0.8	0.883
Birth Weight (g)	4201.5 ± 249.1	3325.9 ± 340.1	<0.001	3331.1 ± 349.9	3283.5 ± 339.9	0.532
Number of pregnancies N (%)			<0.001			0.295
1	153 (37.8)	2209 (57.3)		1979 (57.4)	230 (56.7)	
2	149 (36.8)	1121 (29.1)		1002 (29.0)	119 (29.3)	
≥3	103(25.4)	525(13.6)		469(13.6)	56 (13.8)	
Parity N (%)			<0.001			0.347
1	291 (71.9)	3577 (92.8)		3198 (92.7)	379 (93.5)	
≥2	114 (28.1)	278 (7.2)		252 (7.3)	26 (6.4)	
Interspinal Diameter (cm)	25.4 ± 1.2	25.6 ± 1.9	0.374	25.5 ± 1.8	25.3 ± 1.7	0.115
Intercristal Diameter (cm)	28.2 ± 1.3	28.0 ± 2.2	0.005	28.1 ± 1.8	28.0 ± 2.4	0.236
Sacral External Diameter (cm)	19.8 ± 0.6	20.1 ± 1.4	0.050	19.9 ± 1.2	20.0 ± 1.2	0.434
Transverse Outlet (cm)	8.5 ± 0.2	8.5 ± 0.5	0.971	8.5 ± 0.3	8.5 ± 0.4	0.867

BMI: body mass index.

**Table 2 ijerph-19-03245-t002:** The comparison of the random forest, logistic regression model, and B-ultrasound in the prediction of macrosomia.

Evaluating Indicator	Random Forest	Logistic Regression Model	Ultrasound
Validity			
Sensitivity (%)	91.7	56.2	29.6
Specificity (%)	91.7	82.6	97.6
False-negative rate (%)	8.3	43.8	70.1
False-positive rate (%)	8.3	17.4	2.4
Youden’s index (%)	83.4	38.8	27.2
Predictive value			
Positive predictive value (%)	91.7	70.8	57.7
Negative predictive value (%)	91.7	37.9	93.1

## Data Availability

The data that support the findings of this study are available from the corresponding author upon reasonable request.

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
