# Peer review of "Effective Macrosomia Prediction Using Random Forest Algorithm"

_ijerph, 2022, doi:10.3390/ijerph19063245_

Round 1

Reviewer 1 Report

Thanks for recommending me as a reviewer. In this paper, the authors aimed to develop a new predictive model for recognizing giant bodies using random forests to improve the sensitivity and specificity of giant body prediction. If authors complete major revisions, the quality of the study will be further improved.

  1. The introduction section is well written. However, the introductory section is too short for readers to understand the theoretical background. It will be helpful for readers to understand if the authors describe in more detail the trends of previous studies related to macrosomia prediction and the reasons for using machine learning in the introduction section.
  2. line 54-76: If the authors describe the data source more specifically in the Methods section, it may be helpful to the reader's understanding. For example, authors can be more specific about sampling, etc.
  3. line 83-84: I can't understand this sentence. Please check if it is a typo.  "The subjects in case and control were randomly divided into the training set and the test set according to the percentage of 7:3.17"
  4. Modeling with random forest alone is meaningless. In this study, in order to understand the performance of the predictive model based on the random forest, there must be a comparable model.
  5. The characteristics of the variables used in this study should be more specifically described. Where possible, authors may present in tables.
  6. Authors should be more specific about their findings. Why is Ultrasound's AUC only 0.636?
  7. This study is imbalanced data. Nevertheless, why is the predictive performance of Random Forest so high? How did you deal with imbalanced data?
  8. Authors should add limitations to the discussion section.

Author Response

Point-by-point responses to the reviewers’ comments

Dear Editor,

Thank you for your letter and giving us the opportunity to revise the manuscript ijerph-1557097 entitled “Effective macrosomia prediction using random forest algorithm”. We would like to thank the editors and the reviewers for their valuable comments and suggestions that have greatly improved the quality of this manuscript. Now, we have revised our manuscript in response to the reviewers’ comments and provided explanations, point by point, for each revision request or comment. In order to facilitate the review, we addressed these issues in the “Track Changes”.

We deeply hope our responses are satisfactory. If you have further questions, please let us know by E-mail. Thank you again for your consideration and we await a favorable response to the revision.

Sincerely,

Zhiping Wang

Re: Manuscript Number: ijerph-1557097

Title: Effective macrosomia prediction using random forest algorithm

The responds to the reviewer’s comments:

Reviewer #1:

Comments: Thanks for recommending me as a reviewer. In this paper, the authors aimed to develop a new predictive model for recognizing giant bodies using random forests to improve the sensitivity and specificity of giant body prediction. If authors complete major revisions, the quality of the study will be further improved.

R: Thank you very much for your kindly comments on our manuscript. There is no doubt that these comments are valuable and very helpful for revising and improving our manuscript. In what follows, we would like to answer the questions you mentioned and give detailed account of the changes made to the original manuscript. We will try our best to appreciate your comments and revise carefully to improve the quality of the article.

  1. The introduction section is well written. However, the introductory section is too short for readers to understand the theoretical background. It will be helpful for readers to understand if the authors describe in more detail the trends of previous studies related to macrosomia prediction and the reasons for using machine learning in the introduction section.

R: Thank you for your valuable suggestion. We have supplemented the introduction according to the suggestions. Please refer to line 28-71 on page 1-2 in the revised manuscript.

  1. line 54-76: If the authors describe the data source more specifically in the Methods section, it may be helpful to the reader's understanding. For example, authors can be more specific about sampling, etc.

R: Thank you for your valuable suggestion. We have revised and supplemented the method according to the suggestions. Please refer to line 82-109 on page 2 in the revised manuscript.

3.line 83-84: I can't understand this sentence. Please check if it is a typo.  "The subjects in case and control were randomly divided into the training set and the test set according to the percentage of 7:3.17"

R: Thank you for your valuable suggestion. We have corrected the sentence, please refer to line 133-134 on page 3. There was an error while importing the reference, "17" after "7:3.17" is the reference number. Thank you for reminding us that we have neglected this problem. We have changed 17 to the reference, and the sampling ratio is 7:3, please refer to line 139 on page 3 in the revised manuscript.

4.Modeling with random forest alone is meaningless. In this study, in order to understand the performance of the predictive model based on the random forest, there must be a comparable model.

R: Thank you for your valuable suggestion. We added logistic regression model in the revised manuscript. Please refer to line 140-146 in method section on page 3, refer to line 245-247 in outcome section on page 4, refer to line 439-441 in discussion section on page 9 in the revised manuscript.

5.The characteristics of the variables used in this study should be more specifically described. Where possible, authors may present in tables.

R: Thank you for your valuable suggestion. According to your opinions, the characteristics of gestational age and birth weight were added in Table 1 on page 6 in the revised manuscript.

6.Authors should be more specific about their findings. Why is Ultrasound's AUC only 0.636?

R: Thank you for your valuable suggestion. We have added some details of our findings, please refer to line 238-242 on page 4 in the revised manuscript.

7.This study is imbalanced data. Nevertheless, why is the predictive performance of Random Forest so high? How did you deal with imbalanced data?

R: Thank you for your valuable suggestion. Firstly, the predictive performance of Random Forest is better than many traditional statistical models, which is the reason why it is increasingly used in the medical field. One study used pre-pregnancy and 26-week follow-up data by building random forest model to predict large for gestational age, with an AUC of 0.824[1]. In our study, using the same data for prediction, the AUC of logistics regression is 0.720, while that of random forest can reach 0.953. So the predictive performance has something to do with the methodology of the model. In additional, there may be a situation of imbalance data. In order to minimize the impact of imbalance data on the result, we have compared the 405 people who are selected with those who are not selected. The results showed that 405 people were representative of the unselected population. Please refer to Table 1. And the modified under-sampling method was chosen to make the samples equitably represented.

Reference

  1. Kuhle, S., et al., Comparison of logistic regression with machine learning methods for the prediction of fetal growth abnormalities: a retrospective cohort study. Bmc Pregnancy and Childbirth, 2018. 18.

8.Authors should add limitations to the discussion section.

R: Thank you for your valuable suggestion. We added limitations in the discussion section. Please refer to 465-469 on page 9 in the revised manuscript.

Thanks to the reviewer and the editors for their efforts in improving the quality of papers.

Reviewer 2 Report

Report on “Effective macrosomia prediction using random forest algorithm” by Fangyi Wang et al.

The manuscript investigates the application of random forest model to predict macrosomia. Authors used traditional variables such as age, pre-pregnancy BMI, number of pregnancies, and parity in combination with extrapelvic measurement information in the twelfth week of pregnancy, including interspinal diameter, intercristal diameter, sacral external diameter, and transverse outlet. I have serious concerns about the methodology used to validate the performance of the developed model.

Concerns:

  • The criterion used to define macrosomia was birthweight > 4000 gr, but no reference was made to another criteria as percentile birthweight >90th which is related to gestational age at birth. In this respect, there is a lack of information in the Table 1, gestational age at birth, birthweight or gender.
  • The authors declare that controls are randomly selected, but no matching procedure was used for this purpose? Propensity score or whatever, please clarify.
  • The information about random forest building must be completed, how many variables are used in the build of each tree (%)?, and how many data?
  • The authors used a set of traditional variables with limited ability to predict macrosomia, but they reached a high discrimination ability. For me it is not clear which data were used to validated data. Data was split in 70%-30%, therefore the AUC has to be obtained with the 810*0.3=243 validation data, please confirms because the AUC seems to be overoptimistic and probably estimated with all data (training+validation).
  • Unfortunately, there is agreement about the limited predicted ability of ultrasound, but the time interval between ultrasound and delivery is a crucial factor to increase this ability, for example, at a 35th week of gestational age the AUC must be greater. Please clarify.
  • In the comparison with ultrasound, the authors inform about the 405 macrosomia cases, but again, the comparison must be performed based only on the validation data. A random forest can be overfitting, only external data can be used for comparison purposes.
  • The ROC curves plotted in figure 3 corresponds to dichotomic variables, this is why appears two lines in the graph. The proposed random forest provides a continuous variable, the ROC curve cannot be as the plotted in Figure 3. Similarly, the projected weights obtained by Hadlock formula is a continuous marker, the ROC curve cannot be two lines. Please clarify.

Author Response

Point-by-point responses to the reviewers’ comments

Dear Editor,

Thank you for your letter and giving us the opportunity to revise the manuscript ijerph-1557097 entitled “Effective macrosomia prediction using random forest algorithm”. We would like to thank the editors and the reviewers for their valuable comments and suggestions that have greatly improved the quality of this manuscript. Now, we have revised our manuscript in response to the reviewers’ comments and provided explanations, point by point, for each revision request or comment. In order to facilitate the review, we addressed these issues in the “Track Changes”.

We deeply hope our responses are satisfactory. If you have further questions, please let us know by E-mail. Thank you again for your consideration and we await a favorable response to the revision.

Sincerely,

Zhiping Wang

Re: Manuscript Number: ijerph-1557097

Title: Effective macrosomia prediction using random forest algorithm

The responds to the reviewer’s comments:

Reviewer #2:

Comments: The manuscript investigates the application of random forest model to predict macrosomia. Authors used traditional variables such as age, pre-pregnancy BMI, number of pregnancies, and parity in combination with extrapelvic measurement information in the twelfth week of pregnancy, including interspinal diameter, intercristal diameter, sacral external diameter, and transverse outlet. I have serious concerns about the methodology used to validate the performance of the developed model.

R: Thank you very much for your attention to the model. There is no doubt that these comments are valuable and very helpful for revising and improving our manuscript. In what follows, we would like to answer the questions you mentioned and give detailed account of the changes made to the original manuscript.

1.The criterion used to define macrosomia was birthweight > 4000 gr, but no reference was made to another criteria as percentile birthweight >90th which is related to gestational age at birth. In this respect, there is a lack of information in the Table 1, gestational age at birth, birthweight or gender.

R: Thank you very much for your kindly comments on our manuscript. Macrosomia refers to the diagnostic criteria for term-birth with birth weight ≥4000g in China. Therefore, gestational age of term-birth newborns was not considered in this paper. Gestational age and birth weight of two groups were added in Table 1. Please refer to Table 1 on page 6 of the revised manuscript.

2.The authors declare that controls are randomly selected, but no matching procedure was used for this purpose? Propensity score or whatever, please clarify.

R: Thank you very much for your kindly comments on our manuscript. The method of sampling was supplemented in the method section. Please refer to the line 93-109 on page 2 in the revised manuscript.

3.The information about random forest building must be completed, how many variables are used in the build of each tree (%)?, and how many data?

R: Thank you very much for your kindly comments on our manuscript. We added the information in the outcome section according to your suggestions. Please refer to the line 218-235 on page 3 in the revised manuscript.

4.The authors used a set of traditional variables with limited ability to predict macrosomia, but they reached a high discrimination ability. For me it is not clear which data were used to validated data. Data was split in 70%-30%, therefore the AUC has to be obtained with the 810*0.3=243 validation data, please confirms because the AUC seems to be overoptimistic and probably estimated with all data (training+validation).

R: Thank you very much for your kindly comments on our manuscript. In the process of random forest modeling, many studies have set training-sets accounting for 70% and test-sets accounting for 30%. [1]Before modeling, the total data sets are divided into two independent data sets. In our study, the data were 568 in the training-set and 242 in the test-set. The two data sets are operated independently.

Reference

  1. Sun, H., et al., Prediction of arrhythmia after intervention in children with atrial septal defect based on random forest. Bmc Pediatrics, 2021. 21(1).

5.Unfortunately, there is agreement about the limited predicted ability of ultrasound, but the time interval between ultrasound and delivery is a crucial factor to increase this ability, for example, at a 35th week of gestational age the AUC must be greater. Please clarify.

R: Thank you very much for your kindly comments on our manuscript. We used the last B-ultrasound before delivery for prediction, and the AUC of B-ultrasound in our study was only 0.636, the low discrimination may be related to the close to delivery. This limitation has been added in the discussion. Please refer to the line 430-433 on the page 9 in the revised manuscript.

6.In the comparison with ultrasound, the authors inform about the 405 macrosomia cases, but again, the comparison must be performed based only on the validation data. A random forest can be overfitting, only external data can be used for comparison purposes.

R: Thank you very much for your kindly comments on our manuscript. Yes, we used the training-set to train the random forest model, and then brought the model into the test-set for verification as you mentioned.

7.The ROC curves plotted in figure 3 corresponds to dichotomic variables, this is why appears two lines in the graph. The proposed random forest provides a continuous variable, the ROC curve cannot be as the plotted in Figure 3. Similarly, the projected weights obtained by Hadlock formula is a continuous marker, the ROC curve cannot be two lines. Please clarify.

R: Thank you very much for your kindly comments on our manuscript. We corrected the ROC curve of the random forest model according to your suggestion and supplemented the logistic regression results in figure 4 on page 7. In addition, as for B-ultrasound, we set it as a dichotomous variable, so we did not modify it.

Thanks to the reviewer and the editors for their efforts in improving the quality of papers.

Reviewer 3 Report

The manuscript proposes a random forest model to predict macrosomia using standard prenatal examination data without the use of ultrasonic diagnosis. The article is interesting however the statistical analysis could be improved.

The authors have used under-sampling to balance the case-control data. There is a significant loss of information in this analysis. Out of the 3855 control sample size they have just use one subsample of 405 leaving out the rest of 3450. The analysis could be repeated at least 9 time with different set of sub-samples and results could be summarised across these data-sets. Similarly, it is not clear if there is a cross-validation of the 70:30 split of the selected sample.

There are no details about the random forest algorithm. The R-packages used for all the analyses should be mentioned. It would be good to provide the R-script for the random forest analysis as a supplementary data.

How does the random forest compare with the simple logistic regression model? It would be good assess the performance of random forest model with other traditional or simple models.

The authors should discuss the clinic utility of the results. How can a trained model be used in practice? The performance of the model in an independent dataset could be examined.

It seems there are several other methods to estimate fetal weight apart from Hadlock formula (reference should be added for this method).

It would be interesting to examine the performance of the random forest model with additional variables from the ultrasonic diagnosis.  For example, the variables used in Hadlock formula can be used in the random forest algorithm which may improve the predictive accuracy.

The authors discuss about the normality test. The results of the normality test should be presented in supplementary text. It is not clear which section this test is relevant.

The authors compare the variables within the control sample in table 1. Note that the full sample (3855) includes the sub-sample and hence the test is not valid. The comparison can be done excluding the sub-sample.

The ROC curve, I think, is drawn based on the final single confusion matrix. This could be done including the cross-validation results and the resampling of the controls mentioned before.

Author Response

Point-by-point responses to the reviewers’ comments

Dear Editor,

Thank you for your letter and giving us the opportunity to revise the manuscript ijerph-1557097 entitled “Effective macrosomia prediction using random forest algorithm”. We would like to thank the editors and the reviewers for their valuable comments and suggestions that have greatly improved the quality of this manuscript. Now, we have revised our manuscript in response to the reviewers’ comments and provided explanations, point by point, for each revision request or comment. In order to facilitate the review, we addressed these issues in the “Track Changes”.

We deeply hope our responses are satisfactory. If you have further questions, please let us know by E-mail. Thank you again for your consideration and we await a favorable response to the revision.

Sincerely,

Zhiping Wang

Re: Manuscript Number: ijerph-1557097

Title: Effective macrosomia prediction using random forest algorithm

The responds to the reviewer’s comments:

Reviewer #3:

Comments: The manuscript proposes a random forest model to predict macrosomia using standard prenatal examination data without the use of ultrasonic diagnosis. The article is interesting however the statistical analysis could be improved.

R: Thank you for your approval of our manuscript. There is no doubt that these comments are valuable and very helpful for revising and improving our manuscript. In what follows, we would like to answer the questions you mentioned and give detailed account of the changes made to the original manuscript.

1.The authors have used under-sampling to balance the case-control data. There is a significant loss of information in this analysis. Out of the 3855 control sample size they have just use one subsample of 405 leaving out the rest of 3450. The analysis could be repeated at least 9 time with different set of sub-samples and results could be summarised across these data-sets. Similarly, it is not clear if there is a cross-validation of the 70:30 split of the selected sample.

R: Thank you very much for your kindly comments on our manuscript. We have modified it according to your suggestion. The results for the 9 times with different set of sub-samples was supplemented in figure 5 to perform Cross-validation. Please refer to figure 5 on page 8 in the revised manuscript.

2.There are no details about the random forest algorithm. The R-packages used for all the analyses should be mentioned. It would be good to provide the R-script for the random forest analysis as a supplementary data.

R: Thank you very much for your kindly comments on our manuscript. We have modified it according to your suggestion. Please refer to line 167-173 on page 3 in the revised manuscript.

3.How does the random forest compare with the simple logistic regression model? It would be good assess the performance of random forest model with other traditional or simple models.

R: Thank you very much for your kindly comments on our manuscript. We added logistic regression model in the revised manuscript. Please refer to line 140-146 in method section on page 3, refer to line 245-247 in outcome section on page 4, refer to line 439-441 in discussion section on page 9 in the revised manuscript.

4.The authors should discuss the clinic utility of the results. How can a trained model be used in practice? The performance of the model in an independent dataset could be examined.

R: Thank you very much for your kindly comments on our manuscript. We have added it in discussion. Please refer to line 469-482 on page 9 in the revised manuscript.

5.It seems there are several other methods to estimate fetal weight apart from Hadlock formula (reference should be added for this method).

R: Thank you very much for your kindly comments on our manuscript. We have added it in discussion. Please refer to line 419-425 on the page 9 in the revised manuscript.

  1. It would be interesting to examine the performance of the random forest model with additional variables from the ultrasonic diagnosis.  For example, the variables used in Hadlock formula can be used in the random forest algorithm which may improve the predictive accuracy.

R: Thank you very much for your kindly comments on our manuscript. Our model is used to predict macrosomia at 12 weeks of gestation, and we cannot obtain additional variables from the ultrasonic diagnosis at 12 weeks of gestation, so these variables cannot be included in the model.

7.The authors discuss about the normality test. The results of the normality test should be presented in supplementary text. It is not clear which section this test is relevant.

R: Thank you very much for your kindly comments on our manuscript. We have deleted that part in the revised manuscript.

8.The authors compare the variables within the control sample in table 1. Note that the full sample (3855) includes the sub-sample and hence the test is not valid. The comparison can be done excluding the sub-sample.

R: Thank you very much for your kindly comments on our manuscript. We have modified it according to your suggestion. Please refer to table 1 on the page 6 in the revised manuscript.

9.The ROC curve, I think, is drawn based on the final single confusion matrix. This could be done including the cross-validation results and the resampling of the controls mentioned before.

R: Thank you very much for your kindly comments on our manuscript. We have modified it according to your suggestion. Please refer to figure4 and figure 5 on page 7 in the revised manuscript.

Thanks to the reviewer and the editors for their efforts in improving the quality of papers.

Round 2

Reviewer 1 Report

The authors have faithfully completed the revision.

Author Response

Dear Editor,

Thank you for your letter and giving us the opportunity to revise the manuscript ijerph-1557097 entitled “Effective macrosomia prediction using random forest algorithm” again. We would like to thank the editors and the reviewers for their valuable comments and suggestions that have greatly improved the quality of this manuscript. Now, we have revised our manuscript again in response to the reviewers’ comments and provided explanations, point by point, for each revision request or comment. In order to facilitate the review, we addressed these issues in the “Track Changes”.

We deeply hope our responses are satisfactory. If you have further questions, please let us know by E-mail. Thank you again for your consideration and we await a favorable response to the revision.

Sincerely,

Zhiping Wang

Reviewer 2 Report

I appreciate the effort that authors have done to improve the manuscript.My concerns about the building and validation of the model have been clarified, I only have minor concerns about the presentation of the ultrasound results.

The results of the B-ultrasound have been treated as a dichotomic variable, then there is no cut-off point to apply for this model, results are 0/1. The ROC curve is a catologue of cut-off points and the measure of sensitivity and specificity for these points, therefore the ROC curve has no sense if you have only one cut-off point, you must inform only using the pair of (sensitivity, specificity). Please remove the ROC curve and AUC value for B-ultrasound, informs only by (sensitivity, specificity).

Additionally, as I require in the last revision the authors must inform about the median and interquartile range of gestational age at which the last ultrasound was performed, as they did with the gestational age at birht in descriptive table.

Best regards

Author Response

Dear Editor,

Thank you for your letter and giving us the opportunity to revise the manuscript ijerph-1557097 entitled “Effective macrosomia prediction using random forest algorithm” again. We would like to thank the editors and the reviewers for their valuable comments and suggestions that have greatly improved the quality of this manuscript. Now, we have revised our manuscript again in response to the reviewers’ comments and provided explanations, point by point, for each revision request or comment. In order to facilitate the review, we addressed these issues in the “Track Changes”.

We deeply hope our responses are satisfactory. If you have further questions, please let us know by E-mail. Thank you again for your consideration and we await a favorable response to the revision.

Sincerely,

Zhiping Wang

Re: Manuscript Number: ijerph-1557097

Title: Effective macrosomia prediction using random forest algorithm

The responds to the reviewer’s comments:

Reviewer #2:

Comments: I appreciate the effort that authors have done to improve the manuscript. My concerns about the building and validation of the model have been clarified, I only have minor concerns about the presentation of the ultrasound results.

R: Thank you very much for your attention to the ultrasound results. There is no doubt that these comments are valuable and very helpful for revising and improving our manuscript. In what follows, we would like to answer the questions you mentioned and give detailed account of the changes made to the original manuscript.

  1. The results of the B-ultrasound have been treated as a dichotomic variable, then there is no cut-off point to apply for this model, results are 0/1. The ROC curve is a catalogue of cut-off points and the measure of sensitivity and specificity for these points; therefore, the ROC curve has no sense if you have only one cut-off point, you must inform only using the pair of (sensitivity, specificity). Please remove the ROC curve and AUC value for B-ultrasound, informs only by (sensitivity, specificity).

R: Thank you very much for your kindly comments on our revised manuscript. We have deleted the content about B-ultrasound ROC curve in the article. Please refer to the line187-190 on page 4, figure 4 and table 2 in the revised manuscript.

  1. Additionally, as I require in the last revision the authors must inform about the median and interquartile range of gestational age at which the last ultrasound was performed, as they did with the gestational age at birth in descriptive table.

R: Thank you very much for your kindly comments on our revised manuscript. We have added the content of gestational age at which the last ultrasound was performed. Please refer to the line 176-179 on page 4 in the revised manuscript.

Thanks to the reviewer and the editors for their efforts in improving the quality of papers.

Reviewer 3 Report

The authors have responded to my comments. 

Author Response

(The authors gave the same response as above.)
